# A Pilot Study of Fluorescence-Guided Resection of Pituitary Adenomas with Chlorin e6 Photosensitizer

**DOI:** 10.3390/bioengineering9020052

**Published:** 2022-01-28

**Authors:** Elizaveta I. Kozlikina, Kanamat T. Efendiev, Andrey Yu. Grigoriev, Olesia Y. Bogdanova, Igor S. Trifonov, Vladimir V. Krylov, Victor B. Loschenov

**Affiliations:** 1Prokhorov General Physics Institute of the Russian Academy of Sciences, 119991 Moscow, Russia; efendiev.kt@nsc.gpi.ru (K.T.E.); loschenov@nsc.gpi.ru (V.B.L.); 2Institute for Physics and Engineering in Biomedicine, National Research Nuclear University MEPhI, 115409 Moscow, Russia; 3Federal State Budgetary Educational Institution of Higher Education “A.I. Evdokimov Moscow State University of Medicine and Dentistry”, The Ministry of Healthcare of the Russian Federation, 127473 Moscow, Russia; medway@list.ru (A.Y.G.); Ansher-iork1@yandex.ru (O.Y.B.); dr.trifonov@mail.ru (I.S.T.); krylov@neurosklif.ru (V.V.K.); 4The National Medical Research Centre for Endocrinology, 117292 Moscow, Russia

**Keywords:** fluorescence diagnostics, pituitary adenoma, photosensitizer, chlorin e6

## Abstract

Fluorescence diagnostics is one of the promising methods for intraoperative detection of brain tumor boundaries and helps in maximizing the extent of resection. This paper presents the results of a pilot study on the first use of the chlorin e6 photosensitizer and a two-channel video system for fluorescence-guided resection of pituitary adenomas. The study’s clinical part involved two patients diagnosed with hormonally inactive pituitary macroadenomas and one patient with a hormonally active one. All neoplasms had different sizes and growth patterns. The data showed accumulation of chlorin e6 in tumor tissues in high concentrations: Patient 1: 2 mg/kg, Patient 2: 5 mg/kg, and Patient 3: 4 mg/kg. For Patient 1, the residual part of the tumor was not resected since it was intimately attached to the anterior genu of the internal carotid artery. For Patients 2 and 3, no regions of increased Ce6 accumulation were detected in the tumor foci after resection. Therefore, the use of the Ce6 and a two-channel video system helped to achieve a high degree of tumor resection in each case.

## 1. Introduction

Pituitary adenomas (PAs) are benign neoplasms and account for 10–15% of all intracranial primary brain neoplasms [1].

In recent years, endoscopic transnasal transsphenoidal access has become the gold standard in the surgical treatment of patients with PAs, almost completely replacing transcranial and transnasal approaches performed with a microscope [2,3]. The extent of resection of both hormonally active and inactive tumors is a key criterion for a patient’s recovery and determination of prognosis of disease remission [4]. Two of the main points complicating radical tumor resection are its infiltrative growth pattern and its intraoperative ability to determine areas of tumor infiltration [5]. At the moment, to increase the completeness of adenomas’ resection, there exist such techniques as dissection of the dura mater (DM), coagulation, and resection of DM parts with tumor infiltration. However, the problems of intraoperative determination of the neoplasm boundaries with the presence of infiltrative growth in the surrounding structures remain unsolved.

In recent years, new intraoperative navigation methods, including fluorescence diagnostics (FD), have been applied in neurosurgical practice [6,7,8,9]. FD is based on the administration of photosensitive molecules, photosensitizers (PS) to the patient, mainly accumulating in tumor tissue, with further excitation by a laser light source with a wavelength in the range of the maximum absorption spectrum of PS [10].

Registration of the PS fluorescence allows to more accurately determine the neoplasm boundaries, as well as residual foci of fluorescence as the main part of the tumor is removed [8,11].

There are several PS that are approved by the FDA for clinical use that have various accumulation mechanisms—cellular or vascular [12]. In several studies on fluorescence-guided resection of PAs, 5-aminolevulinic acid (5-ALA)-induced protoporphyrin IX (PpIX) was used [13,14]. PpIX accumulates in tumor cells due to the increased activity of enzymes in the initial stage of heme synthesis in tumor cells, as well as a deficiency in ferrochelatase, an enzyme that utilizes PpIX by converting it into heme. The accumulation of PpIX in tumor cells occurs within a few hours, while in healthy cells, PpIX is converted into a non-phototoxic heme [15]. However, the absence or weak specificity of the 5-ALA-induced PpIX for this type of tumor has been shown [13].

At the moment, indocyanine green (ICG) is mainly used as a PS for fluorescence-guided resection of PAs [16,17,18]. It is a water-soluble dye that binds to albumin when administered intravenously. ICG mainly absorbs between 600 and 900 nm and emits fluorescence between 750 and 950 nm [19]. This wavelength range lies in the optical tissue transparency window, which provides a greater probing depth of pathological tissues. However, ICG has a short half-life [19], and therefore fluorescent monitoring during the entire surgical process is impossible.

Chlorin e6 (Ce6) and its derivatives are promising PSs with high diagnostic and therapeutic efficiency in neuro-oncology [20,21,22]. Ce6 is a second-generation PS with high efficacy and minimal dark toxicity [23]. Chlorin-type PSs have two peaks in the absorption spectrum located in the red and blue bands of the visible spectrum and intense fluorescence in the 640–700 nm range [24]. Ce6 accumulates mainly in blood cells and blood vessels. However, the Ce6 circulating through the blood vessels is able to easily diffuse into the tumor tissue due to the enhanced permeability and retention effect [25]. At the moment, there is no data on the possibility of using this type of drug for fluorescence-guided resection of PAs. In addition to FD, Ce6 can be used for photodynamic therapy (PDT) of residual parts of the tumor that are impossible to resect.

In connection with the above, the aim of this work was to evaluate the possibility of clinical application of chlorin-type PS and a two-channel video-fluorescent system for FD of PAs. The main tasks were to determine the degree of the PS accumulation in tumor tissues and to assess the extent of PAs resection using this technology.

## 2. Materials and Methods

### 2.1. Two-Channel Video System

A two-channel video-fluorescence system (Biospec, Moscow, Russia) consisting of a white light source, a laser source with a radiation wavelength of 635 nm wavelength, a system of optical filters, and a rigid endoscope with a 45° tilt of the distal edge was used for conducting FD (Figure 1). This system allows receiving video data in three modes: color, black and white (B/W), and overlay mode. Visualization in the overlay mode occurs by superimposing B/W and color flow with the coloring of fluorescence foci in green for visual assessment of pathology boundaries. This system makes it possible to evaluate the change in the fluorescence index registered in the zone of interest relative to tissue considered to be healthy [26]. The DM lining the anterior wall of the sella turcica was used as conditionally healthy tissue. Video-fluorescence diagnostics allowed identifying areas with increased PS accumulation, which is typical for pathologically altered tissues.

The metrological characteristics of the system were determined on optical phantoms of pituitary tissue with Ce6 in various concentrations (0, 0.5, 1, 5 mg/kg) and a scattering medium (1% Intralipid MLT/LST) registered with the same calibration as in the analysis of tumor tissue (Figure 2a).

### 2.2. Photosensitizer Ce6

As part of the study, a chlorin-type PS was used to perform FD of PAs, the active substance of which is the trisodium salt of chloride e6 (Fotoditazin^®^, DEKO Company, Moscow, Russia). Ce6 is a second-generation PS widely used in clinical practice, and is characterized by rapid excretion from the body [27]. Additionally, Ce6 has low dark phototoxicity, a long lifetime of the triplet state, and a high quantum yield of singlet oxygen [28,29].

The maximum contrast of Ce6 accumulation in tumor tissues was observed 3–4 h after intravenous administration. PS is excreted from the body or metabolized within the first 48 h after administration [30]. This PS has intense absorption peaks at wavelengths of 402 and 660–670 nm and intense fluorescence in the red optical range with a maximum at 665 nm (Figure 2b).

### 2.3. Clinical Part

The study’s clinical part involved 3 patients. Patients 1 and 2 were diagnosed with hormonally inactive pituitary macroadenomas, and patient 3 with a hormonally active one. All patients underwent endoscopic transnasal removal of the tumor at A.I. Evdokimov Moscow State University of Medicine and Dentistry. Two patients (1, 2) included in the study had previously undergone surgery with resection of the tumor at different times and had continued tumor growth. After the initial intervention, according to the histological examination, patients were diagnosed with PA, while Patient 3 had a primary PA. Additionally, at the preoperative stage, the infiltrative nature of tumor growth into the surrounding tissues was assumed. All patients were admitted to the clinic in satisfactory condition, without focal neurological symptoms. Patients 1 and 2 of the concomitant diseases had hypertension, and Patient 3 had secondary hypothyroidism. Patients 1 and 2, both in the preoperative period and after the surgery, took hormone replacement therapy for hypopituitarism: hydrocortisone and desmopressin. Patient 3 was taking L-thyroxine for a concomitant disease. None of the patients, either in the preoperative period or after the surgery, took chemotherapy drugs, since at the moment they are not included in the standard of treatment for patients with pituitary adenomas, given the benign nature of these types of tumors. All patients underwent a study of the hormonal profile and MRI of the brain with contrast enhancement with an assessment of the tumor spread and its interaction with the main nerve structures (Figure 3, Figure 4 and Figure 5). The patients’ primary characteristics are presented in Table 1.

### 2.4. Intraoperative Video-Fluorescent Study Plan

Ce6 (Fotoditazin^®^) diluted in 100 mL of saline was intravenously injected to the patients at a 1 mg/kg concentration 3–3.5 h before FD. The tumor was accessed using an optical endoscope and a white light source (Karl Storz, Tuttlingen, Germany). After the DM was opened, the localization, the size of residual foci, the concentration of the PS accumulated in the tumor focus, and the zones of infiltration by tumor tissue with an increased fluorescence index relative to tissue considered to be healthy were determined using a two-channel video-fluorescence system. The intact anterior leaflet of the DM, lining the anterior wall of the sella turcica, was taken as healthy tissue. As the tumor was removed, the data were recorded into video files and images for further processing. Tumor resection was stopped when the residual part of the tumor and its capsule infiltrated by the tumor tissue were intimately fused with vital structures, such as the internal carotid artery and oculomotor nerves.

## 3. Results

### 3.1. Patient 1

After removal of the main endosuprasellar node, sensitized tissue with intense Ce6 fluorescence was found in the projection of the right cavernous sinus using the FD method (Figure 6a). After removal of the visible tumor node from the right cavernous sinus, FD was repeated and a section of the cavernous sinus wall with an increased fluorescence index was found (Figure 6b). This area was intimately attached to the anterior genu of the internal carotid artery and therefore was not removed. In the postoperative period, radiation therapy was recommended to the patient in order to exclude a recurrence.

### 3.2. Patient 2

After removal of tumor tissue from the saddle cavity and suprasellar component, FD of the first cavernous sinus zone, where the tumor spread according to the MRI, was performed. A high fluorescence index was registered in this area (Figure 7a). During the control navigation of the tumor bed, no areas with a high fluorescence index of the PS were detected (Figure 7b).

### 3.3. Patient 3

After removal of the visible part of the tumor node spreading endosupralaterocellarly, FD of the tumor bed was performed. A region with increased fluorescence was detected (Figure 8a). After removal of the residual focus, FD was repeated (Figure 8b). No areas of increased fluorescence were detected.

All patients in the postoperative period felt satisfactory, and there was a positive dynamic of visual disturbances. All patients were discharged on day 3–5 after surgery without complications. Follow-up of the patients was 1.5 months. Chiasmal syndrome regressed in all 3 patients, and Patient 2 partially preserved oculomotor disorders in the form of diplopia. All patients had no signs of nasal liquorrhea or bleeding.

## 4. Discussion

Accurate definitions of the PA boundaries and areas of tumor infiltration of the surrounding tissues are key to a high degree of resection, which can increase the likelihood of successful treatment and preserve normal pituitary functions [31]. The authors of [32] have shown that intraoperative MRI reduces the frequency of incomplete removal of pituitary tumors, but this is an expensive method that increases the duration of surgery and is not available in many clinics. One of the reasons why surgical resection cannot guarantee a successful outcome is the complexity of intraoperative determination of the PA boundaries. During the standard procedure of surgical removal of adenoma tissues using endoscopic equipment, we performed video-fluorescence diagnostics, which allowed us to visualize the foci of Ce6 accumulation and quantify the fluorescence intensity of the studied tissues. As far as we know, Ce6 was used for the first time to visualize the PA tissues and their further removal.

At the moment, the most commonly used PS for FD of PA is ICG with fluorescence in the near-infrared region [16,17,18]

In the following studies, after bolus injection of IGG solution in patients with histologically confirmed PA, unaffected tissues showed a stronger fluorescent ICG signal than adenoma, whereas in our study, after intravenous administration of Ce6, the high selectivity of the PS accumulation in adenoma tissues was observed. The greatest contrast in the case of ICG was observed 0.5–3 min after administration, which complicates the process of tumor tissue resection due to the rapid removal of ICG.

The authors of [13], during a multicenter study of the use of 5-ALA for fluorescence imaging of several pathologies of the brain, obtained very unfavorable results. It is noteworthy that only 1 out of 15 patients with PA showed weak fluorescence; in all other cases, there was no PpIX fluorescence

Therefore, the choice of a PS for complete adenoma resection and the possibility of subsequent PDT of residual foci of the PS accumulation is a difficult task due to the characteristics of normal and affected tissues. It should be taken into account that the permeability of the blood–brain barrier in the pituitary gland is high, so the molecules of Ce6 can quickly diffuse between the vascular and interstitial spaces. There are also significant differences in microvascular density between normal and pathological pituitary tissues [33].

Since Ce6 mainly accumulates in the vascular system of neoplasm tissues, this PS is most often used for PDT. Selective light exposure to the zones of Ce6 accumulation primarily destroys the blood vessels feeding the tumor, since they are more prone to thrombosis. The Ce6 antitumor effect also includes damage to both pathological cells by the mechanism of necrosis and/or apoptosis, and tumor-associated macrophages [34,35]. The use of Ce6 during FD and PDT, according to the latest published data, shows high diagnostic and therapeutic efficacy [36,37,38,39,40,41]. Moreover, Ce6 demonstrates higher therapeutic efficacy and fewer side effects compared to hematoporphyrin-based PS [42]

Normalization of the fluorescence index was performed by assigning the fluorescence index value of 10 to the DM lining, which allowed to intraoperatively identify areas with increased fluorescence intensity of Ce6 in all three patients. The highest value of the fluorescence index for Patient 1 was 62 ± 5 rel.u., for Patient 2—130 ± 5 rel.u., and for Patient 3—94 ± 5 rel.u.

After the adenoma removal, video-fluorescence navigation showed a significant decrease in the fluorescence index in all areas, which characterizes the absence of residual foci and a high degree of resection. The obtained results are very encouraging and indicate a high selectivity of the Ce6 accumulation in the PA tissues, which gives prospects for further PDT of sensitized residual pathological foci of PA.

## 5. Conclusions

This paper presented clinical results of the first use of Ce6 PS and a two-channel video system for fluorescence-guided resection of PAs performed on three patients with different tumor sizes and growth patterns. After a comparative analysis of the fluorescence indexes registered on optical phantoms with Ce6 and data recorded during tumor resection in each case, the accumulation of the PS in pathological tissues was determined: Patient 1: 2 mg/kg, Patient 2: 5 mg/kg, and Patient 3: 4 mg/kg.

In the first case, the residual part of the tumor with an increased fluorescence index was intimately attached to the anterior genu of the internal carotid artery and was not resected. In the remaining parts of the tumor bed, no increased fluorescence signal was observed. For patients 2 and 3 at the end of resection, no regions with Ce6 accumulation were detected.

Since the degree of radicality of tumor resection is of particular importance in hormone-active pituitary adenomas, as well as in recurrent tumors, the routine use of the FD method will increase the number of patients with normalization of hormonal status and remission of the disease. Additionally, a high level of Ce6 accumulation in pituitary adenoma tissues and further fluorescence-guided resection can help to predict the possibility of recurrence of the disease and the need for additional treatment, in particular, PDT in patients with foci of infiltrative growth, accumulating PS, if removal is impossible.

## Figures and Tables

**Figure 1 bioengineering-09-00052-f001:**
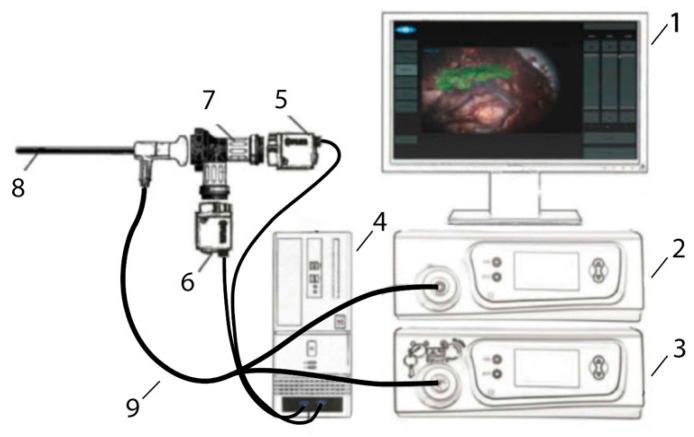
Two-channel video system. 1—Monitor, 2—LED-based white light source with filter, 3—diode laser with filter, 4—camera control unit based on a PC with GPU, 5—B/W camera with filter, 6—color camera with filter, 7—endoscopic adapter with dichroic beam-splitter, 8—regular endoscope, 9—Y-shaped fiber-optic light guide.

**Figure 2 bioengineering-09-00052-f002:**
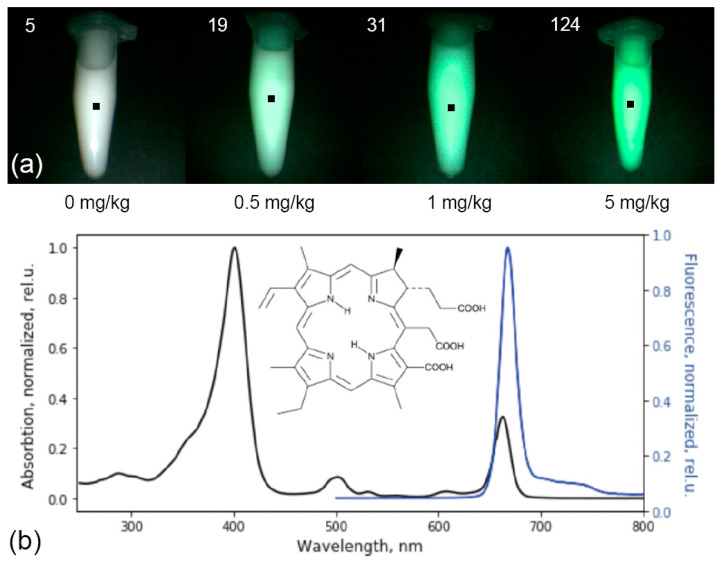
The emitted Ce6 fluorescence in the optical phantom is highlighted in green and matched with the actual image (**a**). Normalized absorption and fluorescence spectra of the Ce6 photosensitizer with its structural formula (**b**).

**Figure 3 bioengineering-09-00052-f003:**
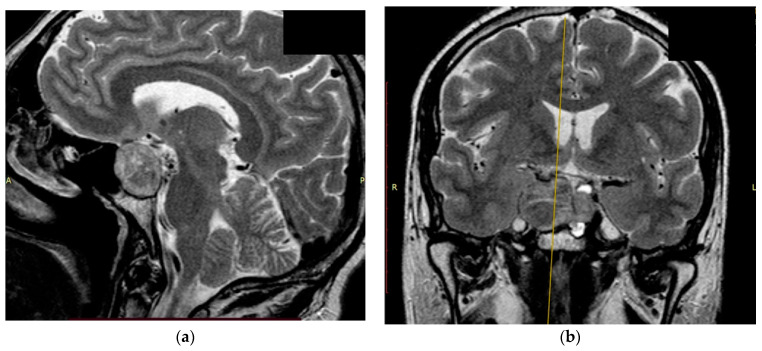
Preoperative T2 MRI images of Patient 1: (**a**) sagittal projection, endosuprasellar mass with chiasm compression, and (**b**) frontal projection, extension to the right cavernous sinus.

**Figure 4 bioengineering-09-00052-f004:**
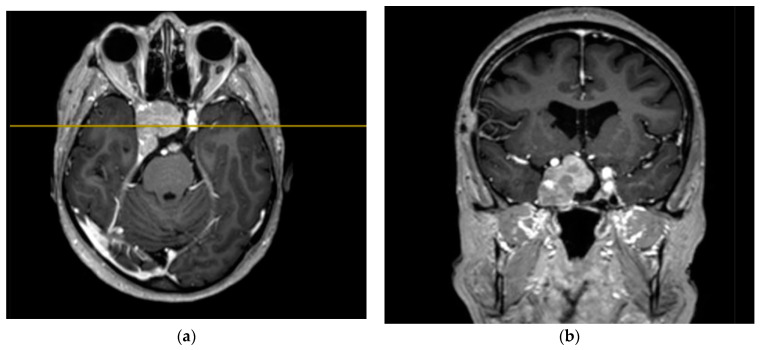
Preoperative T1 MRI images of Patient 2 with contrast enhancement: (**a**) axial projection, endosuppralaterosellar mass with extension to the right cavernous sinus, and (**b**) frontal projection, endosuppralaterosellar mass, with compression of the chiasm, spread to the right cavernous sinus with ingrowth of the right internal carotid artery.

**Figure 5 bioengineering-09-00052-f005:**
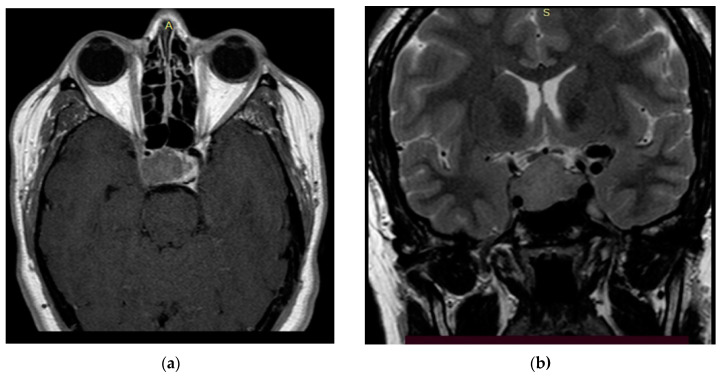
Preoperative MRI images of Patient 3: (**a**) T1 image with contrast, axial projection, supralaterosellar tumor fragment, and (**b**) T2 image, frontal projection, endosuppralaterosellar mass with chiasm compression.

**Figure 6 bioengineering-09-00052-f006:**
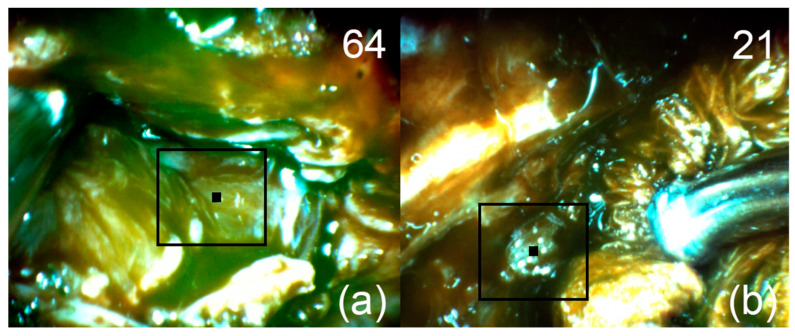
Patient 1. Pictures of fluorescent regions of the tumor (**a**) and residual (**b**) pituitary tissue. The black square corresponds to the zone of interest with accumulated Ce6. The upper right corner shows the fluorescence indexes calculated at the points where the black marker is placed.

**Figure 7 bioengineering-09-00052-f007:**
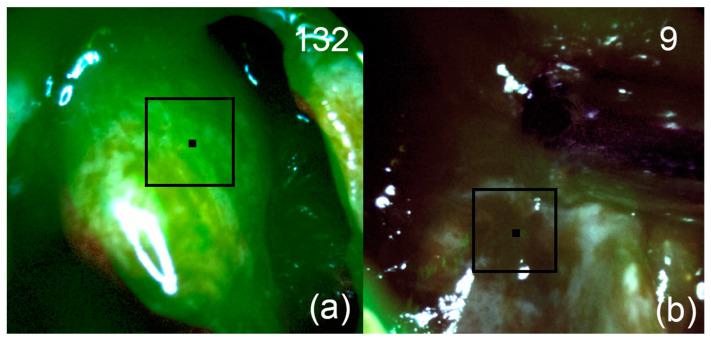
Patient 2. Pictures of fluorescent regions of pituitary tissue before (**a**) and after (**b**) fluorescence-guided resection. The black square corresponds to the zone of interest with accumulated Ce6. The upper right corner shows the fluorescence indexes calculated at the points where the black marker is placed.

**Figure 8 bioengineering-09-00052-f008:**
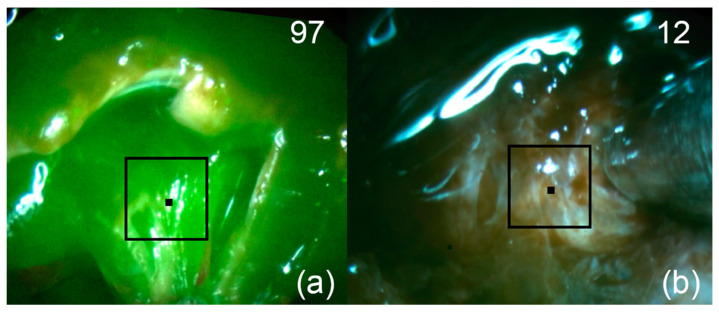
Patient 3. Pictures of fluorescent regions of pituitary tissue before (**a**) and after (**b**) fluorescence-guided resection. The black square corresponds to the zone of interest with accumulated Ce6. The upper right corner shows the fluorescence indexes calculated at the points where the black marker is placed.

**Table 1 bioengineering-09-00052-t001:** Patients participating in the clinical trial.

Patients Data	Patient 1	Patient 2	Patient 3
Gender	M	F	F
Age	51	66	32
Tumor size, mm	22 × 24 × 34	36 × 38 × 32	28 × 32 × 26
Predominant spread of the tumor relative to the sella turcica	endosupralaterosellar		
Main clinical manifestations	chiasmatic syndrome, hypopituitarism	chiasmatic syndrome, oculomotor disorders, hypopituitarism	chiasmatic syndrome
Time since the first surgery, years	12	10	0
KPS ^1^	90	90	100

^1^ Karnofsky performance scale.

## Data Availability

The authors confirm that the data supporting the findings of this study are available within the article.

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
