# Peer review of "A Pilot Study of Fluorescence-Guided Resection of Pituitary Adenomas with Chlorin e6 Photosensitizer"

_bioengineering, 2022, doi:10.3390/bioengineering9020052_

Round 1
Reviewer 1 Report
1) The title should mention the type (e.g. case report/ case series) of the manuscript. The authors should observe that this journal only publishes articles or reviews. This manuscript's best classification is case report/case series. This query should be addressed with the journal’s editor.
2) The abstract should describe more about the cases and the data found. Remember that only the abstract is indexed in the majority of the databases.
3) The authors should better describe why Ce6 was used in these patients. A correlation with a previous study should be done.
4) Please describe the characteristics of each patient. Other diseases, confounding characteristics, medications in use, chemotherapy attempts, why type of adenoma, abnormal laboratory results, etc.
5) Could the authors increase the number of participants in the study?
6) A paragraph discussing the diseases that could influence the levels of Ce6 should be done.
7) How much time the individuals were followed?
Reviewer 2 Report
The Authors presents the use of Ce6 photosensitizer and a two-channel video system during pituitary adenoma surgery. The results, although the small numbers of patients analyzed, are promising and interesting.
Reviewer 3 Report
I applaud the efforts of the authorship team. Well done. The manuscript is sound and provides readers with valuable information. Nice figures to compliment the write-up. Recommend accept.
Round 2
Reviewer 1 Report
1) ‘‘Fluorescence-guided Resection of Pituitary Adenomas with Chlorin e6 Photosensitizer’’
If this article is an ‘‘original article’’, the tile is missing important information about the type of the study and where was done. Also, some journals have a special format for this type of manuscript such as technical notes, protocols, or methods. One of these words is advised to include.
2) OK
3) OK
4) OK
5) Based on the authors' comment about ‘‘Could the authors increase the number of participants in the study?’’. The reviewer did not find in the manuscript text that it was a pilot study. It is advised to include it throughout the text. Also, this can be added to the title.
6) OK
7) OK
Author Response
From: Ms. Elizaveta Kozlikina
Prokhorov General Physics Institute of the Russian Academy of Sciences, Moscow, Russia
National Research Nuclear University MEPhI, Moscow, Russia
Re: Response to reviewer’s comment and suggestion
Dear reviewer,
According to the comments, we've changed the title of the manuscript to "A Pilot Study of Fluorescence-guided Resection of Pituitary Adenomas with Chlorin e6 Photosensitizer".
Also, we've added information to the abstract:
Line 17: This paper presents the results of a pilot study on the first use of chlorin e6 photosensitizer and a two-channel video system for fluorescence-guided resection of pituitary adenomas.